# Copper-Catalyzed Diboron-Mediated *cis*-Semi-Hydrogenation of Alkynes under Facile Conditions

**DOI:** 10.3390/molecules27217213

**Published:** 2022-10-25

**Authors:** Yuxi Zeng, Honggang Zhang, Daofan Ma, Guangwei Wang

**Affiliations:** Department of Chemistry, School of Science, Tianjin University, Tianjin 300354, China

**Keywords:** copper catalysis, *cis*-selective semi-hydrogenation of alkynes, bis(pinacolato)diboron

## Abstract

*Cis*-alkenes are ubiquitous in biological molecules, which makes it greatly significant to develop efficient methods toward construction of *cis*-olefins. Herein, we reported a facile semi-hydrogenation of alkynes to *cis*-alkenes in an efficient way with cuprous bromide/tributylphosphine as the catalyst and bis(pinacolato)diboron/methanol as the hydrogen donor. The method features convenient and facile reaction conditions, wide substrate scope, high yields, and high stereoselectivity.

## 1. Introduction

The *cis*-alkene functionality is widely occurring in pharmaceuticals, fine chemicals, pesticides, and natural products [1,2,3,4,5,6]. For example, *cis*-combretastatin A4, a stilbene derivative from *Combretum caffrum*, is considered to be a strong cell growth and tubulin inhibitor [7]; *cis*-asarone, derived from *Acorus gramineus*, has antifungal activity [8]; cruentaren A, a cytotoxic natural product isolated from myxobacterium *Byssovorax cruenta*, exhibits selective inhibition of F-ATPases, thus showing cytotoxicity against a variety of cancer cell lines [9,10]; chavicine, a *cis*-alkene, is proven to be the precursor compound for the neuroprotective effects of black pepper [11]; haliclonacyclamine F, a bis-piperidine alkaloid, has been isolated from the marine sponge *Pachychalina alcaloidifera* [12] (Figure 1). Therefore, methods for the construction of double bonds with high stereoselectivity have long attracted the interest of the synthetic community, and several related approaches have been developed, such as Wittig reaction [13], Horner–Emmons–Wadsworth reaction [14], Julia–Kocienski reaction [15], Peterson reaction [16], Takai olefination [17], olefin metathesis [18], cross-coupling reaction [19], alkyne semi-hydrogenation [20], halide elimination [21], and so on.

Among these methods for the selective construction of double bonds, alkyne semi-hydrogenation is an attractive route due to its simplicity, atomic economy, and highly controllable stereoselectivity [22,23,24]. Semi-hydrogenation of alkynes by Pb(OAc)_2_-modified Pd/CaCO_3_, widely known as Lindlar reduction, is the first developed alkyne semi-hydrogenation reaction, and has been widely used in total synthesis [25]. In addition to Lindlar catalysts, a variety of homogeneous or heterogeneous catalytic hydrogenation systems for semi-hydrogenation based on Pd [26], Ru [27,28], Rh [29,30], Ir [31], V [32], Nb [33], Co [34], Cr [35], Mn [36], and Fe [37] have been developed (Figure 2a). Nevertheless, the existing Lindlar-type reactions inevitably use high-pressure hydrogen as the hydrogen source, which poses a number of limitations to the reaction, such as potential explosion hazards, cumbersome operations for the use of high-pressure hydrogen, possible over-hydrogenation, and isomerization side reactions. In order to tackle these shortcomings, synthetic scientists developed the transfer hydrogenation reactions, [38,39] which use stable and easily handled reducing agents such as silanes [40,41], formic acid [42], alcohols [43,44], ammonia borane [45,46], DMF [47], hypophosphoric acid [48,49], and amines [50] as indirect hydrogen sources (Figure 2b), avoiding the use of flammable hydrogen gas.

Diboron reagents, which are highly stable and easy to handle, have served as common borylation reagents and are widely used in transition-metal catalyzed borylation reactions [51,52]. Meanwhile, the exploitation of the intrinsically reducing B–B bond has attracted considerable attention due to its obvious advantages in terms of safety and green chemistry compared to the commonly used silanes [53,54,55]. For example, in 2016, Stokes’ group published the transfer hydrogenation of carbon–carbon double bonds catalyzed by Pd/C with B_2_(OH)_4_/water as the reductant in dichloromethane [56]. In 2019, Liu’s group discovered a method for selective transfer of hydrogen from ethanol to alkynes with the assistance of NHC ligands and *^t^*BuOK (Figure 2c) [57]. In the same year, Shi’s group developed a similar method for the *cis*-selective semi-deuteration of alkynes, with the difference that expensive xantphos and LiO*^t^*Bu were used to facilitate the reaction (Figure 2c) [58]. Although these copper-catalyzed semi-hydrogenation of alkynes have made remarkable developments, all of them must use structurally complex and expensive catalytic systems, thus simple and facile reaction systems for this process still require further exploration. Based on the continued exploration of the properties of diboron reagents in our group [59,60,61,62], we herein report a copper-catalyzed alkyne semi-hydrogenation based on B_2_pin_2_-mediated transfer hydrogenation, which requires only simple and cheap *^n^*Bu_3_P and NaOH while good stereoselectivity is highly maintained.

## 2. Results and Discussions

To explore the optimal reaction conditions, diphenylacetylene was chosen as the standard substrate. Through screening of copper catalysts, CuBr was found to be the best catalyst for this semi-hydrogenation (entries 1–4, Table 1). The absence of ligands or addition of other phosphine ligands resulted in lower yields (entries 5–8, Table 1). Poor yields were obtained when B_2_(OH)_4_ was used instead of B_2_pin_2_ (entry 9). Other bases, including LiO*^t^*Bu (entry 10), weak bases (entry 11), and strong organic bases (entry 12), were not as effective as NaOH. Screening of the solvents showed that DMF as solvent provided the best reaction results (entries 13–17, Table 1). Unfortunately, a lower reaction temperature resulted in a decrease in the yield and stereoselectivity (entries 18–19, Table 1). Then, control experiments were conducted which demonstrated that both the copper catalyst and B_2_pin_2_ were indispensable for the reaction (entries 20 and 21). Based on the above screening results, we chose the reaction conditions in entry 1 as the optimal conditions.

Following the optimization of the reaction, a series of alkynes were tested to demonstrate the scope of the reaction (Figure 1). For different internal alkynes, the target products (**2a**–**b**, **2f**–**l**) were obtained in moderate to excellent yields as well as good to excellent stereoselectivity, regardless of the electron-withdrawing or electron-donating group, particularly, the readily reduced carbonyl (**2j**) and cyano groups (**2k**) were compatible with these reaction conditions. Different substitution patterns, including *ortho*-, *meta*-, *para*- and multisubstitution, had slight effect on the efficiency and selectivity of the reaction (**2c**–**e**). The 1-naphthyl-containing substrate gave **2m** in moderate yields, however, with excellent stereoselectivity, which may be attributed to the steric hindrance of the 1-naphthyl group. For the substrate **1n** containing 2-thienyl group, excellent yield and stereoselectivity were obtained. This strategy was also applicable to monoalkyl or dialkyl substituted alkynes (**2o**–**t**) with good to high stereoselectivity, but lower yields resulted for the bulky alkynes. To our delight, the unprotected hydroxyl group did not cause negative effects on the reaction (**2p** and **2r**). Under the above reaction conditions, the semi-hydrogenation products could also be obtained from terminal alkynes (**2u**–**w**). The occasionally lower stereoselectivities observed in the cases of **2c**, **2d**, **2f**, and **2q** have not been reasonably explained yet since the stereoselectivity is the result of combination effects of steric hindrance and electronic effects of functional groups, making it difficult to predict which factor predominates (see SI for a possible isomerization mechanism which involves a reversible addition–elimination process).

## 3. Mechanistic Study

Control experiments were performed to gain further insight into the reaction mechanism. First, isotope labeling experiments (Figure 3a) were carried out using CD_3_OD and anhydrous sodium methoxide. The deuterated product was obtained in 90% deuteration and 83% yield, indicating that the double-bonded hydrogen in the product originated from methanol. Then, the alkenyl boron compound **3** was subjected under standard reaction conditions without B_2_pin_2._ Product **2a** was afforded in 82% yield, which suggests that **3** may be the possible reaction intermediate (Figure 3b).

Based on the above experimental results and the previous literature [56,57,58,59,60,61,62,63,64], we proposed the following possible reaction mechanism as shown in Figure 4. First, the reaction of CuBr with sodium hydroxide and ligand might lead to catalytically active species **I**, which then undergoes a transmetallation reaction with B_2_pin_2_ to give copper-boron complex **II**. The subsequent insertion reaction of alkyne **1a** into complex **II** via species **III** gives *cis*-selective intermediate **IV**. Then, complex **IV** undergoes alcoholysis to generate intermediate **3** while regenerating complex **I**. Further deboration of **3** in the presence of methanol and sodium hydroxide gives *cis*-selective semi-hydrogenation product **2a**.

## 4. Materials and Methods

All experiments were conducted under argon atmosphere. All commercially available reagents were purchased and used without further purification, unless otherwise stated. Flash chromatographic separations were carried out on 200–300 mesh silica gel. Reactions were monitored by TLC and GC analysis of reaction aliquots. GC analysis was performed on an Agilent 7890 gas chromatograph using an HP-5 capillary column (30 m × 0.32 mm, 0.5 μm film) with appropriate hydrocarbons as internal standards. ^1^H, ^13^C, and ^19^F NMR spectra were recorded in CDCl_3_ on a Bruker AVANCE III spectrometer and calibrated using residual undeuterated solvent (CDCl_3_ at 7.26 ppm ^1^H NMR, 77.16 ppm ^13^C NMR). Chemical shifts (δ) are reported in ppm and coupling constants (J) are in Hertz (Hz). The following abbreviations were used to explain the multiplicities: s = singlet, d = doublet, t = triplet, q = quartet, m = multiplet. High resolution spectra (HRMS) were recorded on a QTOF mass analyzer with electrospray ionization (ESI) through a Waters G2-XS QTOF mass spectrometer.

### Experimental Procedures and Characterization of Products

General procedure: To a mixture of **1** (1.0 mmol, 1.0 equiv), CuBr (14.5 mg, 0.1 mmol, 0.1 equiv), ^n^Bu_3_P (50 μL, 0.2 mmol, 0.2 equiv), B_2_pin_2_ (279.3 mg, 1.1 mmol, 1.1 equiv), NaOH (160.0 mg, 4.0 mmol, 4.0 equiv), and MeOH (0.2 mL, 5.0 mmol, 5.0 equiv) was added 8.0 mL of DMF under argon. The reaction mixture was then placed in a preheated oil bath at 80 °C for 12 h. After the reaction was completed, the reaction was diluted with 15 mL of water, then extracted with ethyl acetate (3 × 15 mL). The combined organic extracts were dried over anhydrous MgSO_4_, filtered, and then concentrated. The residue was further purified by silica gel column chromatography to give the semi-hydrogenation product **2**.

(*Z*)-1,2-diphenylethene (**2a**) [65]. 

According to the general procedure on a 0.2 mmol scale, the product was obtained in 98% yield (35.4 mg) as a colorless oil after silica gel column chromatography (PE). R_f_ = 0.58 (PE). ^1^H NMR (400 MHz, CDCl_3_) δ 7.30–7.22 (m, 10H), 6.64 (s, 2H). ^13^C NMR (101 MHz, CDCl_3_) δ 137.4, 130.4, 129.0, 128.3, 127.2.

(*Z*)-1-methyl-4-styrylbenzene (**2b**) [66].

According to the general procedure on a 0.2 mmol scale, the product was obtained in 80% yield (31.1 mg) as a white solid after silica gel column chromatography (PE). M.p. 115.1–116.6 °C. R_f_ = 0.55 (PE). ^1^H NMR (400 MHz, CDCl_3_) δ 7.34–7.21 (m, 5H), 7.18 (d, *J* = 8.0 Hz, 2H), 7.06 (d, *J* = 8.0 Hz, 2H), 6.59 (s, 2H), 2.35 (s, 3H). ^13^C NMR (101 MHz, CDCl_3_) δ 137.6, 137.0, 134.4, 130.3, 129.7, 129.04, 128.97, 128.9, 128.3, 127.1, 21.4.

(*Z*)-1-Methoxy-4-styrylbenzene (**2c**) [26].

According to the general procedure on a 0.2 mmol scale, the product was obtained in 96% yield (40.4 mg, Z/E = 24/1) as a white crystalline solid after silica gel column chromatography (PE/EA = 40/1). M.p. 135.2–137.2 °C. R_f_ = 0.47 (PE/EA = 20/1). ^1^H NMR (400 MHz, CDCl_3_) δ 7.23–7.06 (m, 7H), 6.67 (d, *J* = 8.8 Hz, 2H), 6.44 (d, *J* = 12.3 Hz, 1H), 6.43 (d, *J* = 12.3 Hz, 1H), 3.69 (s, 3H). ^13^C NMR (101 MHz, CDCl_3_) δ 158.8, 137.7, 130.3, 129.9, 129.8, 128.93, 128.86, 128.4, 127.0, 113.7, 55.3.

(*Z*)-1,2-bis(4-methoxyphenyl)ethene (**2d**) [67].

According to the general procedure on a 0.2 mmol scale, the product was obtained in 92% yield (44.2 mg, Z/E = 14/1) as a white crystalline solid after silica gel column chromatography (PE/EA = 40/1). M.p. 34.1–36.1 °C. R_f_ = 0.45 (PE/EA = 20/1). ^1^H NMR (400 MHz, CDCl_3_) δ 7.21 (d, *J* = 8.6 Hz, 4H), 6.78 (d, *J* = 8.6 Hz, 4H), 6.45 (s, 2H), 3.80 (s, 6H). ^13^C NMR (101 MHz, CDCl_3_) δ 158.6, 130.2, 130.1, 128.5, 113.7, 55.3.

(*Z*)-1-methoxy-3-styrylbenzene (**2e**) [68].

According to the general procedure on a 1.0 mmol scale, the product was obtained in 82% yield (172.4 mg) as a yellow oil after silica gel column chromatography (PE/EA = 40/1). R_f_ = 0.56 (PE/EA = 20/1). ^1^H NMR (400 MHz, CDCl_3_) δ 7.28–7.15 (m, 5H), 7.12 (t, *J* = 8.0 Hz, 1H), 6.83 (d, *J* = 7.6 Hz, 1H), 6.77 (s, 1H), 6.73 (d, *J* = 8.2 Hz, 1H), 6.60 (d, *J* = 12.3 Hz, 1H), 6.55 (d, *J* = 12.3 Hz, 1H), 3.61 (s, 3H). ^13^C NMR (101 MHz, CDCl_3_) δ 159.4, 138.6, 137.4, 130.6, 130.2, 129.3, 129.0, 128.3, 127.3, 121.6, 113.8, 113.4, 55.1.

(*Z*)-1-fluoro-2-styrylbenzene (**2f**) [26].

According to the general procedure on a 1.0 mmol scale, the product was obtained in 83% yield (131.6 mg, Z/E = 23/1) as a colorless oil after silica gel column chromatography (PE). R_f_ = 0.51 (PE). ^1^H NMR (400 MHz, CDCl_3_) δ 7.35–7.23 (m, 7H), 7.12 (t, *J* = 9.2 Hz, 1H), 7.00 (t, *J* = 7.5 Hz, 1H), 6.81 (d, *J* = 12.2 Hz, 1H), 6.71 (d, *J* = 12.2 Hz, 1H). ^13^C NMR (101 MHz, CDCl_3_) δ 160.5 (d, *J* = 247.7 Hz), 136.9, 132.3, 130.6 (d, *J* = 3.4 Hz), 129.1 (d, *J* = 8.2 Hz), 128.9, 128.4, 127.5, 125.1 (d, *J* = 14.5 Hz), 123.7 (d, *J* = 3.5 Hz), 122.7 (d, *J* = 3.2 Hz), 115.7 (d, *J* = 21.9 Hz). ^19^F NMR (376 MHz, CDCl_3_) δ −114.77 (m).

(*Z*)-1-ethyl-2-(4-methoxystyryl)benzene (**2g**).

According to the general procedure on a 0.2 mmol scale, the product was obtained in 87% yield (40.0 mg, with 9% over reduction product) as a colorless oil after silica gel column chromatography (PE/EA = 40/1). R_f_ = 0.57 (PE/EA = 20/1). ^1^H NMR (400 MHz, CDCl_3_) δ 7.23 (m, 2H), 7.18 (d, *J* = 7.4 Hz, 1H), 7.11–7.03 (m, 3H), 6.70 (d, *J* = 8.7 Hz, 2H), 6.62 (d, *J* = 12.2 Hz, 1H), 6.56 (d, *J* = 12.2 Hz, 1H), 3.75 (s, 3H), 2.67 (q, *J* = 7.5 Hz, 2H), 1.21 (t, *J* = 7.5 Hz, 3H). ^13^C NMR (101 MHz, CDCl_3_) δ 158.7, 142.4, 137.0, 130.4, 130.0, 129.7, 129.4, 128.6, 127.6, 127.4, 125.9, 113.6, 55.3, 26.8, 15.1. HRMS (ESI) *m*/*z*: [M + H]^+^ Calcd for C_17_H_19_O^+^ 239.1430; found 239.1436.

(*Z*)-1-bromo-4-(4-butylstyryl)benzene (**2h**).

According to the general procedure on a 1.0 mmol scale, the product was obtained in 59% yield (186.0 mg) as a colorless oil after silica gel column chromatography (PE). R_f_ = 0.60 (PE). ^1^H NMR (400 MHz, CDCl_3_) δ 7.35 (d, *J* = 8.4 Hz, 2H), 7.15 (d, *J* = 8.1 Hz, 2H), 7.14 (d, *J* = 8.4 Hz, 2H), 7.06 (d, *J* = 8.1 Hz, 2H), 6.60 (d, *J* = 12.2 Hz, 1H), 6.46 (d, *J* = 12.2 Hz, 1H), 2.58 (t, *J* = 7.6 Hz, 2H), 1.59 (p, *J* = 7.6 Hz, 2H), 1.36 (dq, *J* = 14.6, 7.3 Hz, 2H), 0.94 (t, *J* = 7.3 Hz, 3H). ^13^C NMR (101 MHz, CDCl_3_) δ 142.4, 136.5, 134.2, 131.5, 131.2, 130.6, 128.8, 128.5, 128.3, 120.9, 35.5, 33.6, 22.5, 14.1. HRMS (ESI) *m*/*z*: [M + H]^+^ Calcd for C_18_H_20_Br^+^ 315.0743; found 315.0743.

(*Z*)-1-fluoro-4-styrylbenzene (**2i**) [69].

According to the general procedure on a 1.0 mmol scale, the product was obtained in 72% yield (142.7 mg) as a colorless oil after silica gel column chromatography (PE). R_f_ = 0.51 (PE). ^1^H NMR (400 MHz, CDCl_3_) δ 7.18–7.09 (m, 7H), 6.82 (t, *J* = 8.7 Hz, 2H), 6.51 (d, *J* = 12.2 Hz, 1H), 6.46 (d, *J* = 12.2 Hz, 1H). ^13^C NMR (101 MHz, CDCl_3_) δ 163.2, 160.7, 137.1, 133.30, 133.27, 130.7, 130.6, 130.4, 129.2, 128.9, 128.4, 127.3, 115.4, 115.2. ^19^F NMR (376 MHz, CDCl_3_) δ −114.65.

(*Z*)-4-acetylstilbene (**2j**) [70].

According to the general procedure on a 0.5 mmol scale under 60 °C with CuBr (0.2 equiv) and ^n^Bu_3_P (0.4 equiv), the product was obtained in 59% yield (65.6 mg, with 4.5% over reduction product) as a white solid after silica gel column chromatography (PE/EA = 40/1) M.p. 138.6–140.9 °C. R_f_ = 0.42 (PE/EA = 20/1). ^1^H NMR (400 MHz, CDCl_3_) δ 7.81 (d, *J* = 8.2 Hz, 2H), 7.33 (d, *J* = 8.2 Hz, 2H), 7.27–7.19 (m, 5H), 6.73 (d, *J* = 12.2 Hz, 1H), 6.61 (d, *J* = 12.2 Hz, 1H), 2.57 (s, 3H). ^13^C NMR (101 MHz, CDCl_3_) δ 197.8, 142.4, 136.8, 135.7, 132.6, 129.3, 129.2, 129.0, 128.51, 128.47, 127.7, 26.7.

(*Z*)-4-styrylbenzonitrile (**2k**) [71].

According to the general procedure on a 1.0 mmol scale, the product was obtained in 62% yield (127.2 mg) as a white solid after silica gel column chromatography (PE/EA = 40/1). M.p. 42.4–44.2 °C. R_f_ = 0.51 (PE/EA = 20/1). ^1^H NMR (400 MHz, CDCl_3_) δ 7.40 (d, *J* = 8.2 Hz, 2H), 7.23 (d, *J* = 8.2 Hz, 2H), 7.20–7.14 (m, 3H), 7.13–7.07 (m, 2H), 6.68 (d, *J* = 12.2 Hz, 1H), 6.48 (d, J = 12.2 Hz, 1H). ^13^C NMR (101 MHz, CDCl_3_) δ 142.1, 136.3, 133.4, 132.1, 129.6, 128.9, 128.6, 128.4, 127.9, 119.0, 110.5.

(*Z*)-4-styryl-1,1’-biphenyl (**2l**) [26].

According to the general procedure on a 1.0 mmol scale, the product was obtained in 60% yield (153.8 mg) as a white solid after silica gel column chromatography (PE). M.p. 64.1–66.5 °C. R_f_ = 0.52 (PE). ^1^H NMR (400 MHz, CDCl_3_) δ 7.60 (d, *J* = 7.3 Hz, 2H), 7.48 (d, *J* = 8.3 Hz, 2H), 7.42 (t, *J* = 7.6 Hz, 2H), 7.35 (m, 5H), 7.30–7.19 (m, 3H), 6.64 (d, *J* = 12.5 Hz, 1H), 6.64 (d, *J* = 12.5 Hz, 1H). ^13^C NMR (101 MHz, CDCl_3_) δ 140.7, 139.9, 137.4, 136.3, 130.5, 129.9, 129.5, 129.0, 128.9, 128.4, 127.4, 127.3, 126.99, 126.93.

(*Z*)-1-styrylnaphthalene (**2m**) [71].

According to the general procedure on a 1.0 mmol scale, the product was obtained in 54% yield (123.6 mg) as a colorless oil after silica gel column chromatography (PE). R_f_ = 0.51 (PE). ^1^H NMR (400 MHz, CDCl_3_) δ 8.22–8.15 (m, 1H), 7.99–7.93 (m, 1H), 7.86 (d, *J* = 8.0 Hz, 1H), 7.62–7.53 (m, 2H), 7.50–7.39 (m, 2H), 7.21–7.16 (m, 5H), 7.14 (d, *J* = 12.2 Hz, 1H), 6.93 (d, *J* = 12.2 Hz, 1H). ^13^C NMR (101 MHz, CDCl_3_) δ 136.8, 135.4, 133.8, 132.2, 131.7, 129.2, 128.62, 128.57, 128.2, 127.7, 127.2, 126.6, 126.2, 126.1, 125.8, 125.0. 

(*Z*)-2-styrylthiophene (**2n**) [26].

According to the general procedure on a 1.0 mmol scale, the product was obtained in 89% yield (165.8 mg) as a colorless oil after silica gel column chromatography (PE). R_f_ = 0.57 (PE). ^1^H NMR (600 MHz, CDCl_3_) δ 7.48–7.41 (m, 4H), 7.40–7.36 (m, 1H), 7.15 (d, *J* = 5.1 Hz, 1H), 7.05 (d, *J* = 3.6 Hz, 1H), 6.96 (dd, *J* = 5.1, 3.6 Hz, 1H), 6.78 (d, *J* = 12.0 Hz, 1H), 6.67 (d, *J* = 12.0 Hz, 1H). ^13^C NMR (151 MHz, CDCl_3_) δ 139.9, 137.4, 129.0, 128.9, 128.6, 128.3, 127.6, 126.5, 125.6, 123.5.

(*Z*)-1-(3,3-dimethylbut-1-en-1-yl)-3-methoxybenzene (**2o**).

According to the general procedure on a 0.5 mmol scale, the product was obtained in 45% yield (42.8 mg) as a colorless oil after silica gel column chromatography (PE/EA = 40/1). R_f_ = 0.64 (PE/EA = 20/1). ^1^H NMR (400 MHz, CDCl_3_) δ 7.11 (t, *J* = 7.8 Hz, 1H), 6.76–6.64 (m, 3H), 6.30 (d, *J* = 12.6 Hz, 1H), 5.52 (d, *J* = 12.6 Hz, 1H), 3.73 (s, 3H), 0.92 (s, 9H). ^13^C NMR (101 MHz, CDCl_3_) δ 159.0, 142.8, 141.0, 128.7, 127.0, 121.7, 114.7, 111.8, 55.3, 31.3, 29.9. HRMS (ESI) *m*/*z*: [M + Na]^+^ Calcd for C_13_H_18_ONa^+^ 213.1250; found 213.1259.

(*Z*)-1-styrylcyclohexan-1-ol (**2p**). 

According to the general procedure on a 1.0 mmol scale, the product was obtained in 44% yield (89.0 mg) as a colorless oil after silica gel column chromatography (PE/EA = 6/1; R_f_ = 0.55 (PE/EA = 3/1). ^1^H NMR (600 MHz, CDCl_3_) δ 7.41 (d, *J* = 7.6 Hz, 2H), 7.35–7.29 (m, 2H), 7.24 (t, *J* = 7.4 Hz, 1H), 6.50 (d, *J* = 12.7 Hz, 1H), 5.72 (d, *J* = 12.7 Hz, 1H), 1.69–1.61 (m, 6H), 1.53–1.44 (m, 4H), 1.30 (s, 1H). ^13^C NMR (151 MHz, CDCl_3_) δ 138.93, 137.84, 129.22, 128.84, 128.15, 127.04, 72.96, 39.10, 25.53, 22.17. HRMS (ESI) *m*/*z*: [M + Na]^+^ Calcd for C_14_H_18_ONa^+^ 225.1250; found 225.1252.

(*Z*)-oct-1-en-1-ylbenzene (**2q**) [15].

According to the general procedure on a 1 mmol scale, the product was obtained in 92% yield (173.2 mg, Z/E = 20/1) as a yellow oil after silica gel column chromatography (PE). R_f_ = 0.71 (PE). ^1^H NMR (400 MHz, CDCl_3_) δ 7.36–7.25 (m, 4H), 7.24–7.18 (m, 1H), 6.41 (d, *J* = 12.0 Hz, 1H), 5.67 (dt, *J* = 12.0, 7.3 Hz, 1H), 2.33 (q, *J* = 7.3 Hz, 2H), 1.45 (p, *J* = 6.9 Hz, 2H), 1.36–1.22 (m, 6H), 0.88 (t, *J* = 6.3 Hz, 3H). ^13^C NMR (101 MHz, CDCl_3_) δ 138.0, 133.4, 128.9, 128.8, 128.2, 126.5, 31.9, 30.1, 29.2, 28.8, 22.8, 14.2.

(*Z*)-4-phenylbut-3-en-1-ol (**2r**) [72].

According to the general procedure on a 1.0 mmol scale, the product was obtained in 94% yield (139.3 mg, with 5% over reduction product) as a colorless oil after silica gel column chromatography (PE/EA = 5/1). R_f_ = 0.61 (PE/EA = 3/1). ^1^H NMR (400 MHz, CDCl_3_) δ 7.44–7.31 (m, 4H), 7.29–7.23 (m, 1H), 6.61 (d, *J* = 11.7 Hz, 1H), 5.72 (dt, *J* = 11.7, 7.4 Hz, 1H), 3.76 (t, *J* = 6.5 Hz, 2H), 2.59–2.68 (m, 2H), 1.87 (s, 1H). ^13^C NMR (101 MHz, CDCl_3_) δ 137.3, 131.3, 128.8, 128.4, 128.2, 126.8, 62.3, 32.0.

(*Z*)-((but-2-en-1-yloxy)methyl)benzene (**2s**) [73].

According to the general procedure on a 0.5 mmol scale, the product was obtained in 71% yield (57.6 mg) as a colorless oil after silica gel column chromatography (PE/EA = 40/1). R_f_ = 0.57 (PE/EA = 20/1). ^1^H NMR (400 MHz, CDCl_3_) δ 7.29–7.15 (m, 5H), 5.66–5.48 (m, 2H), 4.43 (s, 2H), 4.00 (d, *J* = 6.1 Hz, 2H), 1.56 (d, *J* = 6.1 Hz, 3H). ^13^C NMR (101 MHz, CDCl_3_) δ 138.5, 128.4, 128.1, 127.9, 127.6, 126.9, 72.2, 65.5, 13.3.

(*Z*)-1,4-bis(benzyloxy)but-2-ene (**2t**) [74].

According to the general procedure on a 1.0 mmol scale, the product was obtained in 91% yield (244.2 mg) as a colorless oil after silica gel column chromatography (PE/EA = 40/1). R_f_ = 0.52 (PE/EA = 20/1). ^1^H NMR (400 MHz, CDCl_3_) δ 7.27–7.14 (m, 10H), 5.76–5.63 (m, 2H), 4.39 (s, 4H), 4.03–3.91 (m, 4H). ^13^C NMR (101 MHz, CDCl_3_) δ 138.2, 129.6, 128.5, 127.8, 127.7, 72.3, 65.8.

1-ethenyl-4-methoxybenzene (**2u**) [75].

According to the general procedure on a 1.0 mmol scale, the product was obtained in 98% yield (131.5 mg) as a colorless oil after silica gel column chromatography (PE). R_f_ = 0.41 (PE). ^1^H NMR (400 MHz, CDCl_3_) δ 7.37 (d, *J* = 8.6 Hz, 2H), 6.88 (d, *J* = 8.6 Hz, 2H), 6.68 (dd, *J* = 17.6, 10.9 Hz, 1H), 5.63 (d, *J* = 17.6 Hz, 1H), 5.14 (d, *J* = 10.9 Hz, 1H), 3.82 (s, 3H). ^13^C NMR (101 MHz, CDCl_3_) δ 159.5, 136.4, 130.6, 127.5, 114.0, 111.7, 55.4.

1-pentyl-4-vinylbenzene (**2v**) [76].

According to the general procedure on a 1.0 mmol scale, the product was obtained in 81% yield (141.2 mg) as a colorless oil after silica gel column chromatography (PE). R_f_ = 0.61 (PE). ^1^H NMR (400 MHz, CDCl_3_) δ 7.40 (d, *J* = 8.1 Hz, 2H), 7.21 (d, *J* = 8.1 Hz, 2H), 6.77 (dd, *J* = 17.6, 10.9 Hz, 1H), 5.78 (dd, *J* = 17.6, 0.9 Hz, 1H), 5.26 (dd, *J* = 10.9, 0.9 Hz, 1H), 2.73–2.62 (t, *J* = 7.6 Hz, 2H), 1.69 (p, *J* = 7.6 Hz, 2H), 1.49–1.34 (m, 4H), 0.98 (t, *J* = 7.0 Hz, 3H). ^13^C NMR (101 MHz, CDCl_3_) δ 142.8, 136.9, 135.2, 128.7, 126.3, 112.9, 35.8, 31.6, 31.3, 22.7, 14.2.

4-vinyl-1,1’-biphenyl (**2w**) [77].

According to the general procedure on a 1.0 mmol scale, the product was obtained in 98% yield (176.7 mg) as a white solid after silica gel column chromatography (PE). M.p. 119.7–120.9 °C. R_f_ = 0.53 (PE). ^1^H NMR (400 MHz, CDCl_3_) δ 7.79–7.74 (m, 2H), 7.73 (d, *J* = 8.3 Hz, 2H), 7.63 (d, *J* = 8.3 Hz, 2H), 7.59 (t, *J* = 7.5 Hz, 2H), 7.54–7.46 (m, 1H), 6.92 (dd, *J* = 17.6, 10.9 Hz, 1H), 5.96 (dd, *J* = 17.6, 0.7 Hz, 1H), 5.44 (dd, *J* = 10.9, 0.7 Hz, 1H). ^13^C NMR (101 MHz, CDCl_3_) δ 140.8, 140.6, 136.7, 136.5, 128.9, 127.4, 127.3, 127.0, 126.8, 113.9 (See Appendix A).

## 5. Conclusions

In conclusion, we developed an efficient semi-hydrogenation of alkynes that yields *Z*-olefins with high stereoselectivity and moderate to high yields. The method features the advantages of convenient and facile reaction conditions, wide substrate scope, and high stereoselectivity. Preliminary mechanistic studies suggest that the *cis*-intermediate **3** generated by insertion might be a critical intermediate in this transformation. Further studies on the mechanism of this strategy and its application in organic synthesis are still in progress.

## Data Availability

The data presented in this study are available in the Appendix A.

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
