# Peer review of "Copper-Catalyzed Diboron-Mediated cis-Semi-Hydrogenation of Alkynes under Facile Conditions"

_molecules, 2022, doi:10.3390/molecules27217213_

Round 1
Reviewer 1 Report
In this manuscript, the author described a Cu-catalyzed cis-selected semi-hydrogenation of alkyne under the facile condition. Although there are many precedents of this chemistry, developing a highly chemo- and stereo-selective transformation under a facile reaction condition is necessary, from a practical application point of view. Under the current condition, most of the transformations gave excellent stereo-selectivity and satisfactory yield. Meanwhile, the background introduction, references, as well as mechanism study and proposal are well presented. In conclusion, this is a very good complementary study for semi-hydrogenation of alkyne. And, this is a high-quality manuscript as well. Therefore, this reviewer in favor of publication.
Minor quibbles:
1. Line 118, B2pin should be “B2pin2”;
2. A gram-scale reaction should be conducted to further evaluate the generality of this transformation;
Reviewer 2 Report
Remarks and comments on the manuscript titled
Copper-catalyzed diboron-mediated cis-semi-hydrogenation of alkynes under facile conditions
by Yuxi Zeng, Honggang Zhang, Daofan Ma, Guangwei Wang
Nice work and clear scientific content on copper‐catalyzed alkyne semi‐hydrogenation based on B2pin2‐mediated transfer hydrogenation using simple and cheap nBu3P and NaOH with high cis-selectivity and rection yield.
Some comments for your consideration:
Figure 1: methyl groups should be specified in the same way (with Me or without Me) in all structural formulas.
The same in Table 2: structures 2c-e, g, o are presented with Me, while in the remaining structural formulas Me is not shown.
Table 2: how to understand “c“ in 2j, 59%c if in the experimental section you write ”According to the general procedure on a 0.5 mmol scale“. If 1 mmol scale requires CuBr (0.1 mmol), nBu3P (0.2 mmol), did you really use a double increase (L96) of these reagents in a 0.5 mmol scale reaction? What determined the use of a temperature of 60 oC in the reaction when the General procedure states 80 oC? L99: Can 2f be described as “obtained in excellent stereoselectivity”? L102-103: “Different substitution patterns, including ortho-, meta-, para- and multi-substitution, have little effect on the efficacy and selectivity of the reaction (2c-e)”. The substitutions for 2c-e are para-, multi- and meta- and meta-substitution appears to outperform para- and multipara-substitutions. L108 and Table 2: Is the stereoselectivity of 2q high? 2l is missing in the Results and Discussions section. L118: Is the yield of 2a 85% or 82% (L120)? L132: Conclusions section should be named as 4. Conclusions and movied behind the Materials and Methods section,
whereas the Materials and Methods section should be numbered as 3.
What can you say about reduced stereoselectivity in 2c, d, f, q cases? L152: …QTOF. L168 and in the entire experimental section: Rf is required for the compounds which were purified by column chromatography. L177 and L184: Is there a difference between PE/EA and PE/EtOAc? L214-215: Please check and correct 2.61-2.55 (t,… L242 and 246: Please, move M.p. from L246 to 64.1-…(L242). L265 and L281: spacings between …9H). 13C… and …3H). 13C… should be included. L288: 2.64 (m, 2H). Is it a multiplet? A range of values should then be specified.
L306: CDCl3 should be specified as CDCl3.
Reviewer 3 Report
Thank you for sending this nice piece of work entitled Copper-catalyzed diboron-mediated cis-semi-hydrogenation of alkynes under facile conditions. Few points should be taken into consideration before the publication of this article
1- The Z/E ratios are shown in table 2, it is better to use the same pattern for products 2c, 2d, 2f, and 2q to be consistent with the other products and to be easier to compare and check the effect of the donating and withdrawn groups in the aromatic moieties.
2- In figure 4, the authors should demonstrate the intermediate that leads to form III in the proposed mechanism
